# Prompt Tuning or Fine-Tuning - Investigating Relational Knowledge in Pre-Trained Language Models

**Leandra Fichtel**                                                            L.FICHTEL@TU-BS.DE
*Institute for Information Systems, TU Braunschweig*

**Jan-Christoph Kalo**                                                         J.C.KALO@VU.NL
*Knowledge Representation and Reasoning Group, VU Amsterdam*

**Wolf-Tilo Balke**                                             BALKE@IFIS.CS.TU-BS.DE
*Institute for Information Systems, TU Braunschweig*

## Abstract

Extracting relational knowledge from large pre-trained language models by a cloze-style sentence serving as a query has shown promising results. In particular, language models can be queried similar to knowledge graphs. The performance of the relational fact extraction task depends significantly on the query sentence, also known under the term *prompt*. Tuning these prompts has shown to increase the precision on standard language models by a maximum of around 12% points. However, usually large amounts of data in the form of existing knowledge graph facts and large text corpora are needed to train the required additional model. In this work, we propose using a completely different approach: Instead of spending resources on training an additional model, we simply perform an adaptive fine-tuning of the pre-trained language model on the standard fill-mask task using a small training dataset of existing facts from a knowledge graph. We investigate the differences between complex prompting techniques and adaptive fine-tuning in an extensive evaluation. Remarkably, adaptive fine-tuning outperforms all baselines, even by using significantly fewer training facts. Additionally, we analyze the transfer learning capabilities of this adapted language model by training on a restricted set of relations to show that even fewer training relations are needed to achieve high knowledge extraction quality.

## 1. Introduction

Recent research has shown that large pre-trained language models, such as BERT [Devlin et al., 2019], being trained on large amounts of natural text, store large amounts of relational knowledge. Several previous works have shown how this stored knowledge can serve as something comparable to a knowledge graph being able to answer simple factual queries [Petroni et al., 2019, 2020], but also for more advanced applications like question answering [Raffel et al., 2020]. In the seminal paper *Language Models as Knowledge Graphs*, Petroni et al. have shown how arbitrary masked language models can be used to answer basic factual queries in cloze-style fashion using the basic fill-mask capability of a language model [Petroni et al., 2019]. Thus, querying the language model, for example, for the birthplace of Albert Einstein would require completing the sentence *Albert Einstein was born in [MASK]*. The partial sentence *was born in* can now be used to retrieve the birthplace of arbitrary persons without any training data from the knowledge graph. This partial sentence is also known under the term *prompt*.

One major difficulty to achieve high precision for this knowledge extraction task is finding good prompts. The original paper by Petroni et al. has chosen manually designed prompts for each relation of a knowledge graph. Follow-up works have developed various techniques for improving the knowledge extraction quality by improving the prompts. Often, even small changes to the prompt may significantly change the result set and therefore the quality of the knowledge extraction task. As an example, the prompt *was born and raised in* achieves significantly better results for extracting facts of the `birthplace` relation, even though its semantics is slightly different. One type of these techniques, called mining-based, is proposing to mine new prompts from natural language text, i.e., from annotated Wikipedia abstracts [Bouraoui et al., 2020, Jiang et al., 2020b]. More recent techniques train explicit additional models on existing triples to optimize the prompt for achieving optimal knowledge extraction results [Shin et al., 2020, Haviv et al., 2021]. In the best case, such a technique improves the knowledge extraction quality of BERT from 31.1% up to 43.3%. However, existing techniques usually need a significant amount of training data in the form of existing knowledge graph triples and a large amount of training time to optimize prompts using complex additional models.

In this work, we show that, instead of this complex additional prompt tuning, a simple adaptive fine-tuning of the pre-trained language model using few training triples from a knowledge graph already does the trick. We introduce *BERTriple*: Its idea is to continue the pre-trained language model's training using masked sentences built from triples of a knowledge graph, such as `Albert Einstein birthplace [MASK]`, together with the correct place of birth `Ulm`. BERTriple achieves superior results without any complex prompting techniques with significantly fewer training triples and less computational time. Concretely, our technique achieves a precision of 48.4% on the LAMA probe in contrast to only 43.3% of existing techniques. Furthermore, we perform an extensive analysis on how much training data is actually needed to still achieve high-quality results.

To further reduce the amount of training data that is needed, we also inspect the transfer learning capabilities of the pre-trained language model for knowledge extraction. Instead of training the model on every relation of a knowledge graph, some relations profit from the training of semantically related relationships. We perform an evaluation of this transfer learning relation by investigating which relations indeed can profit from the training of other relations.

## 2. Related Work

**Language Models as Knowledge Graphs** In [Petroni et al., 2019], it was shown that pre-trained language models can store large amounts of relational world knowledge similar to large knowledge graphs as Wikidata [Vrandečić and Krötzsch, 2014]. The original paper has proposed the LAMA probe to investigate the performance of language models on this task. The LAMA probe on the one hand consists of triples from the Google relation extraction corpus (Google RE), 41 relations from T-Rex, a Wikipedia text corpus annotated with Wikidata triples, 16 relations from ConceptNet, and 305 questions from SQuAD.

Several works have further investigated the potential and the limitations of using language models as knowledge bases. Poerner et al. show that BERT is far from showing perfect results for knowledge extraction but often relies on very simple heuristics [Poerner

et al., 2020]. They show that BERT often is heavily relying on the name of an entity to guess a plausible result. As an example, a query asking for the nationality of a person with an Italian-sounding name, is usually Italian as well. This heuristic may work in most cases but shows that the language model lacks a correct memorization of facts. Since a large proportion of queries can be solved by these simple heuristics in the LAMA probe, Poerner et al. propose a filtered, more difficult version called *LAMA-UHN* which we also evaluate our model on.

Kassner et al. show how well the knowledge extraction from BERT can be fooled by mispriming or by using negated prompts [Kassner and Schütze, 2020]. The results imply that the language model is not very robust to changes and false clues in the prompt. Furthermore, dealing with negation is not possible. These results are interesting to our work because they show that indeed small changes in the prompt can have a large influence on the outcome of the results.

Contrary to Kassner et al. a recent work of Petroni et al. has shown that similar priming effects may be used to improve the knowledge extraction quality from language models [Petroni et al., 2020]. Small text snippets (context paragraphs) from Wikipedia abstracts, containing information about the entity of interest, are retrieved automatically and appended to the cloze-style query. This way the extraction quality is significantly boosted.

Instead of relying on the knowledge that was captured in the pre-training phase, in [Heinzerling and Inui, 2021] the capabilities of a language model directly on Wikidata triples are investigated. Thereby, the authors examine how the number of entities presented in the language model can be increased, how many parameters are needed for storing large knowledge graphs, and how robust these language models are to varying querying prompts. Particularly, the third research question, investigating the robustness of prompts, is related to our work.

Instead of using language models as a knowledge graph for fact extraction as proposed by Petroni et al., Thorne et al. lift this idea on another level by proposing the idea of a Neural Database [Thorne et al., 2020, 2021]. In contrast to previous papers, knowledge is not retrieved from the pre-trained language model, but from textual facts (the database) that are input to the language model together with a query prompt. Furthermore, the language model is fine-tuned to answer different kinds of queries. These involve simple fact extraction queries, over join queries to more complex aggregation queries.

**Mining-based Prompt Tuning**   Since manually designing prompts, similar to Petroni et al., is on the one hand a very cumbersome task and, on the other hand, often does not give good results, automatic mining approaches have been proposed [Bouraoui et al., 2020, Jiang et al., 2020b]. Both approaches show that, given a training corpus of triples and a text corpus with annotated triples similar to T-Rex, can be used to automatically extract better prompts, outperforming the manual approach. The idea of the approach is to mine prompts for each relation from the training data by extracting sentences from the text corpus that contain these relations. The sentences are then ranked by using the fill-mask task on the training triples. Sentences that are well suited for predicting facts from the training dataset correctly are ranked high. For the knowledge extraction task, multiple of

these mined prompts for the same relation can be used together, so that their results can be combined to provide a merged result list.

Overall, these approaches are very runtime intensive, require large amounts of training triples, and an annotated text corpus, while only slightly improving the extraction quality. The precision achieved in [Jiang et al., 2020b] is 34.1%.

**Learning-based Prompt Tuning**   More recently, two learning-based approaches for improving the prompts for the LAMA probe have been presented. The idea of these approaches is to add an additional learning component that optimizes the prompt to achieve better results on the knowledge extraction task directly. BERTese is a module consisting of an additional $BERT_{base}$ model, which rewrites the existing prompts and passes its output to a pre-trained BERT model [Haviv et al., 2021]. A training dataset of existing triples is used to train the rewriter using the fact extraction task as an objective. Overall, BERTese shows an improvement over manually created prompts and mining-based approaches, achieving 38% precision in the LAMA probe.

A related approach is AutoPrompt, a prompt-learning technique that cannot only be used for the fact extraction task, but also for several other prompt-based tasks [Shin et al., 2020]. A manually created prompt for fact extraction is extended by additional 5 or 7 trigger tokens. The choice of these tokens is optimized by a gradient-based search on training triples. AutoPrompt can even outperform BERTese by several points of precision on the LAMA probe, achieving 43% precision.

## 3. Adaptive Fine-Tuning of BERT

In recent times, two basic ideas on how to solve down-stream tasks with large pre-trained language models (i.e., relation extraction, question answering, text classification, summarization) are discussed intensively: prompt tuning and fine-tuning.

With the success of huge language models, the idea of simply using pre-trained language models as they are, and tuning the prompt has become very popular. Particularly GPT-3 has shown impressive results, offering the possibility of solving various tasks in a zero or few-shot fashion [Brown et al., 2020]. Thus, the model does not need to be adapted to a specific task but can be used for multiple tasks as it is. Also, the idea of automatically tuning the prompt, instead of fine-tuning the complete model, has shown promising results [Lester et al., 2021].

For fact extraction from pre-trained language models, as discussed in this work, several automatic prompt tuning techniques have been presented. Instead of solving the task of fact extraction in a zero-shot fashion, the techniques usually require lots of data, for example, to either extract and rank possible prompts [Bouraoui et al., 2020] or to train additional models to rewrite the prompts [Haviv et al., 2021].

A different approach for improving the results of a language model on a downstream task is fine-tuning. Models can be fine-tuned by training on an additional task-specific dataset. However, recent research has shown that models solving tasks on text which are substantially different from the pre-training data perform much worse since the model lacks robustness [Hendrycks et al., 2020]. In such cases, *adaptive fine-tuning* for a specific domain or task may be performed to overcome this problem [Howard and Ruder, 2018]. Adaptive fine-tuning implies continuing the pre-training objective on a more specific domain (often

out of the training data distribution). We believe that particularly the task of fact extraction from a pre-trained language model is substantially different from the original pre-training objective because the queries are restricted to short factual sentences, which only make up a small part of the original training data. Here, domain adaption could therefore lead to substantial improvements in extraction quality.

In this work, we perform an adaptive fine-tuning on BERT to only adjust the model to the triple-data domain in order to improve its performance in cloze-style relational knowledge extraction. We call the adapted version of BERT: *BERTriple*. In contrast to other standard fine-tunings, like training a binary classifier for relation extraction, we reuse the fill-mask task already used in the pre-training of BERT. In our adaptive fine-tuning, we use facts as triples of the form (subject, relation, object) from an existing knowledge graph and so-called prompts for every relation to query the language model. For example, considering the triple (Albert Einstein, place of birth, Ulm) and the prompt `[S] was born in [O]`, where `[S]` and `[O]` are placeholders for the subject and the object of a triple. Thus, a training data point consists of the masked input, e.g. `Albert Einstein was born in [MASK]`, and the desired target, e.g. `Albert Einstein was born in Ulm`, by putting the object token into the [MASK]-token.

Since Haviv et al. discover that small changes of the input like rewriting `tom terriss is a [MASK] by profession` into `tom terriss is the [MASK] by profession` have a crucial influence on the performance of BERT, we introduce so-called *triple prompts*. In contrast to the natural language prompts, for triple prompts no prompt tuning is necessary anymore because they are reduced to the simplest form to present a triple as input for language models. They are built directly from the entities and relation labels by concatenation and are of the form `[S]` <`label of relation`> `[O]`. For example, the triple prompt of the birthplace relation could be `[S]` `place of birth` `[O]`.

## 4. Experiments

Overall, we perform three major experiments to evaluate the idea of adaptive fine-tuning with triples for fact extraction from the language model. Our datasets and our implementation are available on Github [1]. (a) We compare our fine-tuning method to existing methods for fact extraction on the LAMA and LAMA-UHN probe. These methods involve manually created prompts, mining-based prompts, and learning-based prompts. (b) We investigate how prone our training approach is to using only small amounts of training data. (c) To further investigate the relations separately, we perform an evaluation of the transfer learning capabilities.

### 4.1 Prompt Tuning vs. Adaptive Fine-Tuning

**Experimental Setup**   In the first experiment, we evaluate our model compared to four other methods by using the LAMA and LAMA-UHN probe. We use the evaluation metric *precision at one* (P@1) introduced in [Petroni et al., 2019] by averaging over all queries within a relation and then across all 41 relations. All P@1 values for each method in Table 1 use $BERT_{base}$ as the underlying language model. Additionally, in the appendix A,

---

[1] https://github.com/LeandraFichtel/BERTriple

Table 1: P@1 [%] of four baselines and our model BERTriple evaluated with LAMA and LAMA-UHN probe

| Test Dataset | BERT | LPAQA | BERTese | AutoPrompt | **BERTriple** |
|---|---|---|---|---|---|
| LAMA | 31.1 | 34.1 | 38.3 | 43.3 | **48.4** |
| LAMA-UHN | 21.8 | 28.7 | - | - | **39.1** |

we present the results for several other language models. The four baselines are: (a) **BERT** - based on manually created prompts [Petroni et al., 2019] (b) **LPAQA** - based on mining-based prompts minded from Wikipedia sentences [Jiang et al., 2020a] (c) **BERTese** - based on learning-based prompts created by a rewriter [Haviv et al., 2021] (d) **AutoPrompt** - based on learning-based prompts by constructing customized prompts for a specific task automatically [Shin et al., 2020]

For comparability, we use the same training dataset (called *original*) created by Shin et al. to train our model **BERTriple**. This dataset consists of at most 1000 Wikidata triples extracted from the T-REx Wikipedia corpus for each of the 41 relations used in LAMA and LAMA-UHN. If the T-REx dataset does not contain enough triples, the authors add triples of Wikidata for this relation but there are still some relations that have less than 1000 triples. During fine-tuning, we use the triples to query for the object by using the manually created prompts of Petroni et al. Further, Shin et al. make a 80/20 split into a training and a development dataset. Because our method does not require the development set, we make use of all 1000 triples per relation for training. We choose the hyperparameters as recommended in [Devlin et al., 2019]: 3 epochs, Adam optimizer with learning rate of $5e - 5$, batch size of 16, weight decay with strength of 0.01.

**Results**  As shown in Table 1 using the LAMA probe, the mining-based method LPAQA has the lowest performance with 34.1% precision. Learning-based methods as BERTese or AutoPrompt are able to provide slight improvements up to a precision of 43.3%, but our approach BERTriple still offers the best precision with 48.4%. Especially in comparison to AutoPrompt, our model improves precision by 5% points while using the same number of training triples. In comparison to the precision of the initial baseline BERT, BERTriple achieves a major improvement of more than 50% (31.1% → 48.4%).

Evaluating with the more complex LAMA-UHN test dataset, the precision of BERTriple decreases to 39.1% but it still exceeds the precision of BERT and LPAQA. BERTese and AutoPrompt have not been evaluated with LAMA-UHN yet and consequently, no precision values are available. Overall, the adaptive fine-tuning outperforms the baselines on both the LAMA and LAMA-UHN probe significantly.

### 4.2 Different Training Datasets

**Experimental Setup**  In the second experiment, we evaluate, how a smaller amount of training triples during fine-tuning on the LAMA test dataset affects the precision (P@1). Similar to the first experiment, we use the training dataset of Shin et al. as the maximum sample size and then choose random samples. For example, the sample size *100* means

that for each relation we randomly select 100 out of all original training triples. The sample size *1* is included for validation purposes only to make sure that the fine-tuning does not compromise the model. For robustness reasons, we run the fine-tuning three times for every sample size and randomly reselect the triples each time. We plot the mean of the three precision values.

In addition to the manually created prompts of Petroni et al., we create the triple prompts of the form [S] <label of relation> [O] by using the relation labels already stored in Wikidata. Thus, no extra effort has to be made to obtain these triple prompts and they are easily extendable to new relations. In order to be able to analyze the triple prompts with the LAMA probe, we adjust the evaluation so that the triple prompts are used for the triples of both, the test dataset and the training dataset. For comparability, we make sure that the randomly selected triples are the same for the manual and triple prompts.

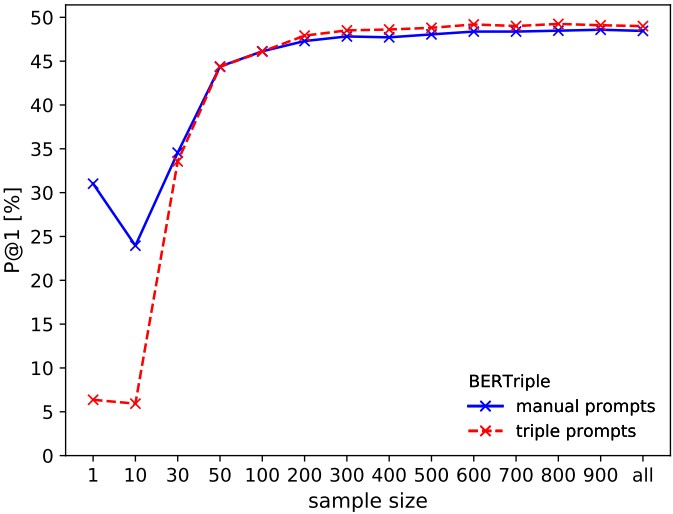

Figure 1: P@1 [%] of our model BERTriple with respect to different sample sizes of training dataset evaluated with LAMA probe

**Runtime** We run our experiments on a single RTX 2080 TI. The fine-tuning with all triples for each relation takes about 45 minutes, whereas, with a sample size of *100*, the training time is only 5 minutes. All in all, our fine-tuning is not demanding with respect to computation time. Exemplary for the other four baselines considered in Section 4.1, we look at the mining-based approach LPAQA by [Jiang et al., 2020a]. Because of the setup of ranking all possible extracted prompts, it is very runtime intensive. For ranking, it is required that for each prompt a set of triples (in this case a maximum of 1000) has to be predicted by BERT using the fill-mask task and this prompt and based on these predictions a ranking score is calculated.

**Results** Figure 1 shows that from a sample size of *50*, state-of-the-art results with a precision of about 44% are achieved. Increasing the sample size, further improves the

precision to about 48% while using all 1000 triples of the training dataset during fine-tuning (see also Table 1). However, it is notable that a sample size of *100* is already sufficient to exceed the precision of Shin et al. Consequently in contrast to the other methods (mining-based and learning-based), adaptive fine-tuning needs significantly less training data to achieve promising results.

The reason for this could be that with adaptive fine-tuning, the knowledge, which is already stored in the weights of the pre-trained model, is used and the weights are only adjusted to the new triple-data domain. As a consequence of the few training triples, our adaptive fine-tuning is very fast and does not require labels since the fill-mask task is reused. Below a sample size of *50*, interesting artifacts are visible. While using 30 random triples during fine-tuning increases the precision, using a sample size of *10* decreases the precision to a level even below the baseline BERT (e.g. manual prompts: $31.1\% \rightarrow 23.95\%$). We believe that 10 triples per relation have enough effect during the fine-tuning to change the weights, but are not enough to represent the data well. Similar behavior is observed in the memorization evaluation in [Heinzerling and Inui, 2021]. There is also a slight decrease in memorization performance if only 5 or 10 triples are used to fine-tune the current prompt variant. Additionally, as we expected, using only 1 triple during fine-tuning does not affect the precision, meaning the precision still matches BERT.

Evaluating the two different prompting techniques (manual prompts and triple prompts), we achieve the same precision values after fine-tuning independently from the prompts that were used. This is remarkable due to the fact that using triple prompts, the baseline BERT does not have a good performance at all (6.37%). This is a valuable result in contrast to BERTese, since using the triple prompts the sentence structure and the choice of words do not have a big impact on the fact extraction performance of BERTripel anymore.

### 4.3 Transfer Learning

**Experimental Setup**  In the third experiment, for every relation, we execute an additional adaptive fine-tuning with the same setup as in Chapter 4.1, but for each fine-tuning the corresponding relation is omitted, meaning there are no triples of this single relation used during training. Thus, all other remaining 40 relations are fine-tuned with 1000 triples each. We evaluate the 41 *omitted models* with the LAMA probe, to investigate the transfer learning capabilities for each relation from the other relations. For this, we compare the precision (P@1) of the baseline BERT, our model BERTriple and of the omitted models for each relation.

**Results**  The results for the experiment are depicted in Table 2. Considering the precision of the omitted models, the relations can be clustered into three groups. The precision is either (a) in the same range of the original BERT, (b) better than BERT and in the same range of our method BERTriple (bold), (c) or notably lower than BERT.

The first group (a) includes, for example, the relations P27 (country of citizenship), P138 (named after) and P937 (work location). P138 achieves a precision of 61.37% with BERT and using the omitted model a precision of 66.98%, which is in the same range. For these relations, no transfer learning occurs if they are omitted during fine-tuning and thus they do not learn from the other relations. They simply revert to the precision they have achieved for pre-trained BERT. Unsurprisingly, there is a little noise compounded onto the

Table 2: P@1 [%] of BERT, BERTriple and BERTriple with omitted relations during training (omitted) to evaluate transfer learning capabilities by using LAMA probe

| ID | Label | BERT | BERTriple | omitted |
|---|---|---|---|---|
| P17 | country | 31.29 | 37.74 | 14.52 |
| P19 | place of birth | 21.08 | 18.86 | 19.81 |
| **P20** | place of death | 27.91 | 32.95 | **31.27** |
| P27 | country of citizenship | 0.00 | 47.41 | 0.41 |
| P30 | continent | 25.44 | 80.41 | 3.59 |
| **P31** | instance of | 36.66 | 69.31 | **44.36** |
| P36 | capital | 62.11 | 57.98 | 53.99 |
| P37 | official language | 54.55 | 63.15 | 55.69 |
| P39 | position held | 7.96 | 43.83 | 11.43 |
| **P47** | shares border with | 13.70 | 16.96 | **15.43** |
| **P101** | field of work | 9.91 | 15.52 | **13.36** |
| **P103** | native language | 72.16 | 88.23 | **79.84** |
| P106 | occupation | 0.63 | 33.40 | 1.98 |
| P108 | employer | 6.79 | 10.18 | 7.83 |
| **P127** | owned by | 34.79 | 51.97 | **38.86** |
| P131 | located in the administrative territorial entity | 23.27 | 35.30 | 25.65 |
| P136 | genre | 0.75 | 64.34 | 7.63 |
| P138 | named after | 61.37 | 75.70 | 66.98 |
| P140 | religion | 0.63 | 76.53 | 4.65 |
| P159 | headquarters location | 32.37 | 36.40 | 31.33 |
| P176 | manufacturer | 85.51 | 89.93 | 84.07 |
| **P178** | developer | 62.94 | 69.88 | **66.67** |
| P190 | twinned administrative body | 2.22 | 3.33 | 1.51 |
| P264 | record label | 9.56 | 50.58 | 13.29 |
| P276 | location | 41.54 | 47.39 | 30.58 |
| **P279** | subclass of | 30.71 | 67.32 | **48.24** |
| **P361** | part of | 23.61 | 48.61 | **36.70** |
| P364 | original language of film or TV show | 44.51 | 53.04 | 44.39 |
| P407 | language of work or name | 64.20 | 71.38 | 32.95 |
| **P413** | position played on team / speciality | 0.53 | 46.95 | **14.50** |
| P449 | original broadcaster | 20.91 | 39.43 | 27.73 |
| P463 | member of | 67.11 | 60.89 | 54.22 |
| P495 | country of origin | 28.71 | 39.60 | 11.22 |
| P527 | has part | 11.17 | 36.07 | 19.36 |
| P530 | diplomatic relation | 2.81 | 3.51 | 3.61 |
| P740 | location of formation | 8.87 | 16.13 | 4.17 |
| P937 | work location | 29.77 | 44.34 | 25.68 |
| **P1001** | applies to jurisdiction | 70.47 | 86.59 | **79.46** |
| P1303 | instrument | 7.59 | 26.55 | 1.37 |
| P1376 | capital of | 73.82 | 51.50 | 63.95 |
| P1412 | languages spoken, written or signed | 65.02 | 77.09 | 60.17 |

precision values of the omitted model. Some values are slightly greater or smaller than the precision achieved at BERT. One reason is that the triples during training are shuffled in each epoch. This noise has no significant effect on the average precision over all relations.

The relations, which belong to the second group (b), are marked in bold. For example, the relation P361 (part of) achieves a precision of 36.70% although there were no training

triples present during fine-tuning. Resulting, it outperforms the original precision of 23.61% with BERT significantly. Thus, for those relations transfer learning can be observed. Some relations (e.g. P20 (place of death) or P178 (developer)) have the same performance with the omitted model and with BERTriple but for most of the bold relations, the performance is slightly lower. This is caused by the fact that a relation is not only learning from its triples alone but often from a selected subset of the relations. So for example in the case of P279 (subclass of), the improvement in precision from BERT to the omitted model (30.71% → 48.24%) is due to the inclusion of a certain subset of the 40 relations during fine-tuning. But the final improvements to achieve the precision of 67.32% are caused by the triples from the relation itself.

The last group (c) is comprised of only a few relations: P30 (continent), P1376 (capital of) and P36 (capital). Evaluating the omitted model, the precision of P30 does not revert to the value achieved with BERT (25.44%) but decreases to 3.59%. The same applies to P36 and P1376 for which the precision with BERTriple falls from 62.11% to 53.98% or respectively from 73.82% to 63.95%. This is an unexpected result since the two relations are inverses of one another and thus they should actually show transfer learning.

Additionally for P36 and P1376, there are interesting artefacts visible: Training all relations already decreases the precision compared to BERT (P36: 62.11% → 57.98%, P1376: 73.82% → 51.50%). For P1376, the precision with BERTriple (51.50%) is even lower than the precision with the omitted model (63.95%). In contrast, there are a couple of relations (e.g. P27 (country of citizenship) or P413 (position played on team)) for which the precision is significantly improved from near zero with BERT to a competitive precision with BERTriple. One reason for these observations could be generalization issues since the weights of the model are adapted during fine-tuning in order to generalize new data for all relations.

## 4.4 Qualitative Evaluation

**Experimental Setup** For this experiment, we have manually performed an analysis of the correct answers of the test queries and the predictions of BERT and BERTriple. We focused particularly on relations with a large difference in precision between BERT and BERTriple as demonstrated in Table 2.

**Results** It is noteworthy that all these relations have in common that the number of distinct object entities is naturally quite small. As an example, the continent relation P30, has only six different possible objects. The predictions of the original BERT model comprise around 80 different answers, mostly consisting of continents, countries, and cities. The most frequent predictions are indeed continents, but often not the correct continent entity. One would expect that our fine-tuning would restrict the variance in these predictions to only continents. However, the range of predicted object entities of BERTriple, similarly to the original BERT, comprises around 80 different entities ranging from continents to cities. But the overall prediction correctness of BERTriple is much higher since the fine-tuning seems to enable the model to better understand the prompt itself. Similar behavior can be observed for the other relations (e.g. P39, P140, P413) that significantly profit from the fine-tuning as well.

## 5. Discussion and Conclusion

In this paper, we have investigated the idea of adaptive fine-tuning a pre-trained language model for cloze-style fact extraction as proposed by [Petroni et al., 2019]. Recent models have shown that the traditional idea of additional prompt tuning can significantly improve the fact extraction performance. Here, either mining-based or learning-based models have been proposed but both techniques share the downside of requiring a lot of training data and complex additional models which have to be trained. Our experiments show that a simple adaptive fine-tuning can significantly improve upon the best state-of-the-art baselines on the LAMA probe, even with fewer training data required and without the need for an additional model. Consequently, the fine-tuning is very fast. Since short *triple prompts* perform as well as manual ones after fine-tuning, using our method, the choice of words no longer has an impact on the performance of the language model, in contrast to the main idea of prompt tuning. We also show that the amount of training data can be reduced even further, due to the interesting transfer learning capabilities of our BERTriple model. Several relations benefit from the fine-tuning with triples of other relations, meaning that they can completely be left out during fine-tuning without affecting their precision significantly.

Comparing prompt tuning techniques with the idea of fine-tuning, there is one main difference. Whereas the main goal of prompt tuning is to use the prompts for many downstream tasks and not to save model checkpoints for each task [Lester et al., 2021], this is no longer the goal of our method. Through the adaptive fine-tuning, our model BERTriple is limited to the cloze-style fact extraction task. For a different task, a new adaptive fine-tuning has to be executed. However, most of the models for prompt tuning are complex and add a significant extra training effort while using these tuned prompts actually results in a worse fact extraction performance in contrast to our adaptive fine-tuning. Consequently, instead of reaching the goal to have a single solution for all tasks, fine-tuning a pre-trained language model offers a more computational efficient solution to achieve superior fact extraction performance. Additionally, our results may open up the possibility to generate simple and well-performing solutions using adaptive fine-tuning for other downstream tasks as well. Even though it requires a change of the original pre-trained language model, adaptive fine-tuning is a very promising method, particularly for cloze-style fact extraction.

**Future Work** While our work shows first promising results for the comparison of prompt tuning and adaptive fine-tuning for relation extraction, we believe that further investigations are needed. Working with larger language models probably leads to different results. Furthermore, current benchmark datasets for cloze-style fact extraction have several limitations: A larger set of relations from Wikidata, which are not extracted from T-REx, should be used, because the LAMA probe is limited only to facts from Wikipedia abstracts. Since real-world knowledge graphs are inherently incomplete, the number of triples per relation is highly skewed, which is not reflected in LAMA. But there are also some natural constraints. For example, for the relation P19 (place of birth) 1,977,898 triples are stored, because every person has exactly one place of birth, but for the relation P1303 (instrument) only 162,634 triples exist, because not everybody has to play an instrument. Therefore, it may be worthwhile to examine which subsets of relations concretely show transfer learning behavior in order to better handle unbalanced data.

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

## Appendix A. Additional Models

Additionally, to the experiments with BERT$_{base}$, we also performe experiments with other masked language models. Concretely, we use BERT$_{large}$, DistilBERT, RoBERTa, and BART. The experimental setup is similar to the one described in Section 4.1. The results of these experiments are presented in Table 3.

**Results**  For the original pre-trained models, very diverse results can be observed. While DistilBERT only achieves a precision of 4.7% on LAMA and only 2.8% on LAMA-UHN, BERT$_{large}$ performs slightly better than BERT$_{base}$.

After adaptive fine-tuning, the quality of all models is increased significantly, with 51.6% for LAMA and 43.1% for LAMA-UHN being the best performance achieved by BERT$_{large}$. BERTriple, BART, and RoBERTa achieve about the same performance on LAMA and LAMA-UHN, while DistilBERT has slightly lower precision.

An interesting observation is the fact that all models, even though they vary in size, architecture, and pre-training corpus, achieve very similar performances after fine-tuning.

Table 3: P@1 [%] for LAMA and LAMA-UHN for several additional language models.

| Model | LAMA | LAMA-UHN |
|---|---|---|
| BERT$_{base}$ | 31.1 | 21.8 |
| BERTriple | 48.4 | 39.1 |
| | | |
| BERT$_{large}$ | 32.3 | 24.4 |
| Fine-Tuned BERT$_{large}$ | 51.6 | 43.1 |
| | | |
| DistilBERT | 4.7 | 2.8 |
| Fine-Tuned DistilBERT | 45.6 | 35.8 |
| | | |
| RoBERTa | 24.7 | 17.0 |
| Fine-Tuned RoBERTa | 48.6 | 39.6 |
| | | |
| BART | 21.1 | 11.5 |
| Fine-Tuned BART | 48.5 | 39.4 |

