# OpenReview forum: "Prompt Tuning or Fine-Tuning - Investigating Relational Knowledge in Pre-Trained Language Models"
_AKBC.ws/2021/Conference — AKBC 2021_

### Official Review · Reviewer_XzDL · 2021-07-15
**Simple but effective method for the task of "fact retrieval" from pre-trained language models, despite modifying the model directly.**

**Rating:** 8
**Confidence:** 4

**Review:**

This paper presents a simple but effective method for the task of "fact retrieval" from pretrained language models. The method requires a training set of  (s,r,o ) triples, which must have human-readable labels, such as wkikdata triples. The triples are treated as pseudo-natural language sentences e.g.  (Dante, born in ,Florence) -> "Dante born in Florence", and the pre-trained language model is given additional MLM finetuning on these training data.
(they also investigate a setting which uses the LAMA templates rather than just directly using triples).

The fine-tuned model is then queried for a fact by providing the subject and relation, and replacing the object with a [mask] token, and then extracting the highest scoring predicted token as the object prediction, similar to cloze-question answering, à la LAMA.

The authors evaluate their approach on LAMA, and show improved P@1 scores, comparing to prompting approaches - both the simple manual approach from the original LAMA probe, and more recent approaches that try to optimize the prompt.
The authors then assess how much training data is required for their approach to be effective, and conclude that about 50 examples per relation is sufficient for the approach to work well.

They also assess to what extent finetuning on triples can improve predictions for unseen relations. They do this by holding out a relation, fine-tuning a model on all the other relations ,and then evaluating the LAMA performance for that relation, and observe that some relations improve relative to the original LAMA probe zero-shot scores, but most are either around the same performance as the original LAMA probe scores or perform very poorly.

The authors conclude with a thoughtful discussion.

This work resembles recent work in "soft prompting", in the sense that explicit prompts are avoided and fine-tuning is used instead, e..g https://arxiv.org/pdf/2101.00190.pdf, https://arxiv.org/abs/2104.08691 and  https://arxiv.org/abs/2103.10385 - the latter paper also experimenting with LAMA and achieving almost the same P@1 for bert-base. Unlike these works however, this model fine-tunes the whole language model, rather than learning only "continuous prompt".

The paper feels similar in spirit to COMET, which also trains a transformer essentially directly on facts from a knowlegebase, although in the case of COMET, its data from commonsense KBs.

Strengths

* the paper is very clear, easy to read and well laid out. The ideas are well-motivated and argued The related work section is a particularly good summary of the important relevant work.
* A directly-finetuned baseline like this is important for contextualizing and understanding other works on this dataset.
* The experiments are thoughtful and well executed
* I particularly like the discussion and future work section, acknowledging the strengths and limitations of this work.
* this approach eliminates the need for manually-written prompts for relations, which is a great strength
* the training process is fast and computationally fairly cheap, and the number of examples needed for the method to work well are quite small
* the authors use both LAMA and its harder subset LAMA-UHN

Weaknesses

* unlike prompting work, this directly fine-tunes the model, which specializes it, and reduces its ability for zero-shot tasks - this is quite a fundamental difference. Whereas other approaches try to fit prompts/queries to the model, this approach changes the model itself - as such, it is unclear wether this is appropriate for a probing dataset like LAMA. The authors are aware, and touch on this in their discussion.
* Only one base model (BERT) is used, and only one dataset is experimented on (LAMA, and its subset LAMA-UHN) - the paper could be improved it it assessed how effective the method was for other models, such as GPT-style models or encoder-decoder models like T5 or BART
* Whilst this approach doesn't require manually written prompt templates for relations, it probably does rely on human-readable relation labels to work well in the low data regime - e..g "born in" rather than a human unreadable label  (e.g. P19) "Q23243". An interesting experiment would be to see how important these are, and wether the method would work with unique but non-human-descriptive relation labels.

Overall, I think this is a well-presented, well written paper giving useful contribution/baseline methods for people interested in fact-retrieval, although i have qualms about treating LAMA as as "task", to be trained an optimized for, rather than a "probe", used for analyzing knowledge already present in LMs.

---

> ### Author Response · Authors · 2021-07-31
> **Additional Models, Experiments and Prompting vs. Fine-tuning**
>
> We would like to thank you for your very comprehensive and helpful review. Your review has given us a new perspective on our work and provided us with interesting related works that we were not aware of.
>
> >the paper could be improved it it assessed how effective the method was for other models, such as GPT-style models or encoder-decoder models like T5 or BART
>
> Indeed, this would be a valuable addition to the paper. Unfortunately, our lab's GPU computing resources are very limited so these additional experiments would take a lot of additional time. We will try to finish experiments for some additional models for the final version of the paper.
>
>
> >An interesting experiment would be to see how important these are, and wether the method would work with unique but non-human-descriptive relation labels.
>
>
> This is an interesting idea and we have performed a first experiment fine-tuning with the property IDs, but keeping the original human-readable entity labels.
> This has worked surprisingly well and we could achieve a performance of just 2% below the human-readable templates when using 100 training triples per property. Due to space reasons, we have not added this experiment to the paper itself, since we believe this would need more space for discussion and analysis.
>
> > Whereas other approaches try to fit prompts/queries to the model, this approach changes the model itself - as such, it is unclear wether this is appropriate for a probing dataset like LAMA.
>
> This seems to be the main criticism of you and another reviewer. We are aware that we modify the original model which might deviate from the original idea of probing for world knowledge. We try to not add additional world knowledge, but to tune the model to better understand the triples. However, we cannot rule out that we add some knowledge to the model.
> We added some more detailed insights and thoughts in the answer to *Reviewer fFyE*. We think the answer might therefore be also interesting for your point.

---

### Official Review · Reviewer_a3cA · 2021-07-21
**Adaptive fine-tuning of pre-trained masked language models for knowledge extraction using masked triple prompts**

**Rating:** 7
**Confidence:** 3

**Review:**

This paper presents the BERTirple model for the cloze-style fact extraction task. Instead of complex additional prompt tuning methods, they show that an adaptive fine-tuning of the pre-trained language model can outperform the best state-of-the-art baselines on the LAMA and LAMA-UHN probes. They propose to use the triples of a knowledge graph in the form of masked sentences to alleviate the difficulties of manually designing prompts. This new data representation leads to better performance, where the proposed model outperforms the previous manually created, mining-based and learning-based prompts. Finally, the experiments show that by omitting some relations during training, the performance doesn't degrade, which validates the transfer learning capabilities of the proposed model.

Strengths:
- The paper is mainly well-written, and the empirical results are promising.
- Their approach doesn’t need to tune the prompts and the idea of using the triples as masked sentences is simple and efficient.
- They show that the proposed model requires fewer training data to outperform the previous SOTA model.
- The discussion about transfer learning experiments reported in the paper is compressive and informative.

Weaknesses:

- In Figure (1), it's better to repeat the experiments (for each sample size) at least 3 times and report the mean and variance of each experiment. Since the paper claims that “by using significantly fewer training facts, BERTriple outperforms all baselines”, the mean and variance of P@1 are required to see the stability of the model.
- Regarding the training speed, authors mention that “existing techniques usually need a significant amount of training data in the form of existing knowledge graph triples and a large amount of training time to optimize prompts using complex additional models” also “ with fewer training data required and without the need for an additional model. Consequently, the fine-tuning is very fast.” Since the training speed is another contribution of the paper, it’s much better to quantify the training speed (and/or fine-tuning) and compare the training time between mining-based, learning-based, and BERTirple.
- It’s a little hard to follow the reported results in Table 2 without the name of each relation. Please add each relation’s name (label) in the final version.

---

> ### Author Response · Authors · 2021-07-31
> **Added insights on robustness and runtime**
>
> Thank you for your helpful comments. We have performed some additional experiments and added some additional information to the paper as stated below:
>
> >In Figure (1), it's better to repeat the experiments (for each sample size) at least 3 times and report the mean and variance of each experiment. Since the paper claims that “by using significantly fewer training facts, BERTriple outperforms all baselines”, the mean and variance of P@1 are required to see the stability of the model.
>
> Thanks for this hint. You are right that particularly the small sample sizes could influence the overall precision of the model. We have retrained the model 3 times for each sample size and now report the mean precision in Figure 1. Since the variance between these 3 runs was very small and hardly depictable in our graphic, we omitted the variance.
>
> >Since the training speed is another contribution of the paper, it’s much better to quantify the training speed (and/or fine-tuning) and compare the training time between mining-based, learning-based, and BERTirple.
>
> We have not retrained all the baseline models ourselves, but used the numbers from the original papers. But our runtime observations stem from several experiments that we did over the last months that have shown us that the other approaches are significantly slower. Our approach, however, for small training samples only needs around 5min of training on a high-end consumer GPU. For further details, we have added a new paragraph in Section 4.2.
>
> >It’s a little hard to follow the reported results in Table 2 without the name of each relation. Please add each relation’s name (label) in the final version.
>
> We have added the labels in Table 2.

---

### Official Review · Reviewer_fFyE · 2021-07-22
**The paper fine-tunes the model directly (instead of performing prompt tuning) and shows strong results on the LAMA benchmark.**

**Rating:** 6
**Confidence:** 4

**Review:**

This paper aims to extract relational facts from pre-trained language models. Instead of tuning the prompts at the input layer of the language model, the paper investigates fine-tuning the model on a set of training facts (<subject, relation, object> triples) for each relation. The results show that fine-tuning the pre-trained language model can achieve higher accuracy on the LAMA benchmark compared to prompt-tuning approaches. The paper also studies the transfer learning capabilities of the pre-trained language models for knowledge extraction and provides interesting insights.

Strengths:
* The paper studies extracting knowledge from pre-trained language models which are trained on unlabeled text. The problem seems promising with the development of modern large-scale language models and it fits well with the scope of AKBC.
* The paper is well written and easy to understand.
* The empirical results on LAMA are strong and significantly outperform previous approaches.
* Transfer learning experiments lead interesting discussions and insights.

Weaknesses:
* The main weakness of this paper is that the goal of this paper is unclear to me. The approach is only evaluated on the LAMA benchmark, but LAMA is originally to probe how much knowledge is stored in the pre-trained language models, where the parameters of the models are frozen. In this paper, the model parameters are updated. If the paper targets to better estimate how much knowledge is stored in the language models, it lacks discussion on whether the results can reflect how much knowledge the model has stored in parameters (because the parameters have been updated). If the paper aims to build a neural knowledge base, the scope of the LAMA benchmark can be limited -- it only contains single-token objects and 41 relations.
* The paper lacks qualitative analyses of the experiments. It would be insightful if the paper can include some concrete examples when showing the experimental results. For example, in transfer learning experiments’ group (c), the “omitted” performance can be much lower than the original BERT. Some examples might help understand what’s happening there.

---

> ### Author Response · Authors · 2021-07-31
> **Probing vs. Fine-tuning and Qualitative Analysis**
>
> First of all, we would like to thank you for your helpful review. We agree that both weaknesses are indeed problematic. We have added additional content, since we got an additional page for the revision.
>
>
> >If the paper targets to better estimate how much knowledge is stored in the language models, it lacks discussion on whether the results can reflect how much knowledge the model has stored in parameters (because the parameters have been updated).
> This point is similar to what the other reviewers have pointed out. Indeed the paper is not being clear about the differences between probing a pre-trained language model for world knowledge and building a neural knowledge base.
>
> We are aware that we modify the original model, which might deviate from the original idea of probing a language model for knowledge. However, the goal was to not add additional world knowledge, but just tune the model to better understand the triples.
> We cannot completely exclude the problem of adding some knowledge to the original model, but similar issues may also exist for pure prompting models like BERTese or AutoPrompt.
> These models might capture additional knowledge during training in the parameters e.g. of the rewriter which then is used for probing the original pre-trained language model. Thus, BERTese for example may learn certain trigger words which it can use in a prompt to trigger a certain answer behavior in the model. E.g. predicting only countries for the object entity when considering P19 (place of birth). This way a prompting model may also capture additional knowledge which complicates the probing procedure and could be considered as task specific fine-tuning without changing the original model.
>
> We believe that this discussion may question the idea of probing a language model for knowledge using complex prompting techniques and would need significantly more investigation of the different techniques. We would be happy to continue this discussion.
>
>
> >The paper lacks qualitative analyses of the experiments. It would be insightful if the paper can include some concrete examples when showing the experimental results. For example, in transfer learning experiments’ group (c), the “omitted” performance can be much lower than the original BERT. Some examples might help understand what’s happening there.
>
> We were able to add an additional qualitative analysis of our experiments in Section 4.4. Here, we particularly focus on properties that have improved significantly from the fine-tuning and added some examples to better understand the effects of BERTriple.

---

### Decision · Program_Chairs · 2021-08-17

**Decision:**

Accept

**Comment:**

This paper proposes a simple method for extracting relational information out of language models: fine-tune the model to fill in masked out entity names, without any prompting (and hence without any prompt tuning). This outperforms baselines that do have to rely on prompt tuning. The reviewers agreed that this paper is strong and well-written.